# Relative Contribution of Metal Content and Soil Particle Mass to Health Risk of Chromium-Contaminated Soil

**DOI:** 10.3390/ijerph19095253

**Published:** 2022-04-26

**Authors:** Shuting Huang, Fei Huang, Xiaojun Yang, Rongbo Xiao, Yunze Wang, Meili Xu, Yuxuan Huang, Hangyuan Shi, Peng Wang

**Affiliations:** 1Guangdong Industrial Contaminated Site Remediation Technology and Equipment Engineering Research Center, School of Environmental Science and Engineering, Guangdong University of Technology, Guangzhou 510006, China; hstngu@126.com (S.H.); xuml0916@163.com (M.X.); hyx_yuxuan720@163.com (Y.H.); shyxiansheng@163.com (H.S.); wangp@jnu.edu.cn (P.W.); 2Department of Geography, Florida State University, Tallahassee, FL 32306, USA; xyang@fsu.edu; 3Guangzhou Nanyang International School, Guangzhou 510000, China; ellie_wang2022@163.com

**Keywords:** particle size, particle mass, bioaccessibility, risk assessment, Cr, contribution

## Abstract

Three soil samples from a chromium (Cr)-contaminated field were classified into five particle fractions (i.e., 0–50 μm, 50–100 μm, 100–250 μm, 250–500 μm, and 500–1000 μm) and were further characterized to study their physicochemical properties and Cr bioaccessibility. The results indicated that the gastrointestinal bioaccessibility estimated by the Solubility Bioaccessibility Research Consortium (SBRC) method was on average 15.9% higher than that by the physiologically based extraction test (PBET) method. The health risk of all samples was within the safe range, and the health risk based on total Cr content may be overestimated by an average of 13.2 times compared to the bioaccessibility-based health risk. The health risk investigated from metal content was mainly contributed by the 50–250 μm fraction, which was 47.5, 50.2, and 43.5% for low-, medium-, and high-level polluted soils, respectively. As for the combined effect, the fractions of 100–250 μm and 500–1000 μm contributed the highest proportion to health risk, which was 57.1, 62.1, and 64.4% for low-level, medium-level, and high-level polluted soils, respectively. These results may further deepen the understanding of health risk assessment and quantify the contribution of the soil particle mass to health risk.

## 1. Introduction

Chromium (Cr) is a category 1 carcinogen, which can pose a serious threat to the ecological environment and living organisms because of its accumulation, carcinogenicity, teratogenicity, and ability to easily bioaccumulate in the human body through the food chain [1]. Trivalent chromium (Cr(III)) and hexavalent chromium (Cr(VI)) are the most common and stable states of Cr in soils [2]. Compared with Cr(III), Cr(VI) can cause significant harm to the environment due to its higher toxicity and mobility [2]. Respiratory inhalation, skin contact, or oral ingestion of Cr(VI) can cause lung cancer, skin allergies, and other symptoms [3,4]. In addition, exposure to high-dose Cr(VI) can lead to renal tubular necrosis and renal failure and also affect the liver, gastrointestinal tract, and cardiovascular system [4]. In China, approximately 600 million tons of chromium slag (2.57% total Cr content) caused by the production process of chromium salt and 31 unowned historical plots polluted by chromium slag needed urgent treatment and restoration, since they exposed local residents to health hazards due to the elevated chromium content. Health risk assessment of soil heavy metals has become an important measure for local governments to control environmental quality and estimate the health risk of residents.

In 2016, the Technical Review Working Group (TRWG) of the USEPA recommended using a particle size smaller than 150 μm to assess the risk of soil exposure to human health [5], and the Bioaccessibility Research Group of Europe (BARGE) recommended that soil fractions smaller than 250 μm be used to assess the oral bioaccessibility of toxic elements [6]. Many researchers have used less than 100-mesh (150 μm) soil particle samples according to national standards to simulate the health risks of pollutants to the human body [5]. Specifically speaking, Ikegami et al. [7] reported that nearly 90% of soil particles on playgrounds attached to children’s hands are smaller than 100 μm. Other studies have suggested using particles smaller than 45 μm to study human risk assessment [8,9]. Fine soil particles are easier to be attached to human skin compared with coarse soil particles, and most of the particles adhering to the hands are smaller than 10 μm [10,11]. Moreover, researchers tend to use in vitro extraction methods simulating lung or gastrointestinal fluid rather than exposing humans to particles for health risk investigation purposes [12,13,14,15]. The ingestible dust particles from soils may be much finer than the soil particles attached to the skin [10]. The risk of particulate matter contamination to human health can be affected by the solubility of pollutants, especially in the digestive and respiratory systems [12,14,16]. Additionally, the total concentration and potential risk of potentially toxic metals in urban soils can be strongly affected by the particle size fraction [6,17,18,19]. A study by Rieuwerts et al. [20] revealed that physicochemical properties, such as pH, OM, and CEC, may affect soil metal content and its bioaccessibility. Despite previous works, there is a lack of quantitative information on the relative contributions of different particle size fractions to total human health risk.

The purpose of this study was (i) to quantify the physicochemical properties and Cr bioaccessibility distribution of samples with different particle size ranges; (ii) to explore the influence of particle size and physicochemical properties on the concentration and bioaccessibility distribution of Cr; and (iii) to quantify the combined effect of metal content and particle mass in different fractions on health risks based on total Cr content and Cr bioaccessibility.

## 2. Materials and Methods

### 2.1. Study Area and Sample Collection

The surface soil samples (0–20 cm) were collected from a major chromate production area in a contaminated field that needed to be remediated for future reutilization in Guangdong Province, China. The field is contaminated by chromium, which may pose potential health risks to nearby residents. Considering the production function area and its concentration difference, three soil samples were collected, representing low (L), medium (M), and high (H) Cr content, respectively. At each sampling point, five subsamples were thoroughly mixed to obtain a composite soil sample (approximately 5.0–7.5 kg) according to the standard method “The Technical Specification for Soil Environmental Monitoring (HJ/T 166-2004)” [21]. The soil samples were then stored in sealed polyethylene bags for further processing.

### 2.2. Sample Particle Size Separation

The samples were air-dried at room temperature, and large pieces of debris and plant residues were removed. Dry samples were sieved through a 2000 μm nylon sieve and then stored in a polyethylene sealed bag for further particle size separation and analyses [18]. According to the texture classification of soil [22] and the study of Ma et al. [17], our study divided the soil samples into coarse sand (500–1000 μm), medium sand (250–500 μm), fine sand (100–250 μm), coarse silt (50–100 μm), and silt and below (0–50 μm). Firstly, the dry sieving method (YLK, ZS-200-8411, Changsha, China) was used to separate the soil particle size through a motorized sieving device. The stacked sieves were shaken at 1400 rpm for 10 min, and then the fractions retained in each sieve were collected. After that, the wet sieving method according to Li et al. [23] was used to sieve soil particles smaller than 50 μm. A 100 g amount of dry soil sample (0–100 μm) was mixed with 1000 mL of deionized water. After dispersing the sample with an ultrasonic probe, the suspension was passed through a 300-mesh nylon sieve (50 μm). After separation, the sample on the sieve was 50–100 μm in size, and the sample in the filtrate was the part smaller than 50 μm. After centrifuging the filtrate at 4000 rpm for 20 min, the supernatant was removed, and the precipitate (0–50 μm) was freeze-dried and weighed for later use.

### 2.3. Sample Physicochemical Properties

Soil samples were characterized for their pH, cation exchange capacity (CEC), and organic matter (OM) content. Soil pH was measured in a proportion of 1:2.5 (soil– deionized water) using a pH meter (INESA, PHS-3C, Shanghai, China) according to the standard method “Determination of pH—Potentiometry (HJ 962–2018)” [24]. CEC content was measured using the standard method “Determination of Cation Exchange Capacity (CEC)—Hexamminecobalt Trichloride Solution—Spectrophotometric Method (HJ 889–2017)” [25]. OM content was measured using the standard method “Determination of Organic Carbon—Potassium Dichromate Oxidation Spectrophotometric Method (HJ 615–2011)” [26]. Total iron content (Fe) was extracted by hydrochloric acid and was measured using an ultraviolet–visible spectrophotometer (METASH, UV-5200, Shanghai, China) after color development by 1,10-phenanthroline [27]. The chemical reagents were of analytical grade, and all experiments were performed in triplicate.

### 2.4. Sample Total Cr Concentration

The total Cr concentration of all samples was digested using the HNO_3_−HCl−HF−HClO_4_ system according to the standard methods “Digestion of Total Metal Elements—Microwave Assisted Acid Digestion Method (HJ 832–2017)” [28]. After the digestion (CEM, MARS6, America), the solution was brought to volume, which was filtered with a 0.45 μm filter and then analyzed. The total Cr concentration was measured using a flame atomic absorption spectrophotometer (HITACHI, Z2000, Kyoto, Japan).

The accumulation factor (AF) can be used to indicate the relative accumulation of Cr in different particle size fractions. Cr tends to accumulate in a specific fraction if AF is more than 1. The accumulation factor (AF) was calculated according to Equation (1).
(1)$$\mathrm{AF}=\frac{C_{\mathrm{different}\;\mathrm{fraction}}}{C_{\mathrm{total}\;\mathrm{concentration}}}$$
where C_different fraction_ is Cr in different particle size fractions (mg/kg). C_total concentration_ is the Cr concentration in the bulk soil calculated after the weighted average of the Cr content and mass of different particle size fractions (mg/kg).

### 2.5. In Vitro Bioaccessibility Tests

The oral bioaccessibility of Cr in soil samples was evaluated using the physiologically based extraction test (PBET) and Solubility Bioaccessibility Research Consortium (SBRC) methods [29,30,31]. With a solid–liquid ratio of 1:100, a 0.30 g sample of each fraction was mixed with 30 mL of gastric phase solution, and the pH of the solution was adjusted to 2.50 ± 0.05 and 1.50 ± 0.05 with hydrochloric acid for the PBET method and SBRC method, respectively. The mixture was shaken in a programmable incubator shaker (LONGYUE, DJ5-2013R-2, Shanghai, China) at 37 °C and at a speed of 200 rpm for one hour. After extraction, the sample was centrifuged at 4000 rpm for 5 min (Cence, H2050R, Changsha, China), and the supernatant was passed through a 0.22 μm filter membrane and stored at 4 °C for testing. The pH of the remaining solution was adjusted to 7.00 ± 0.05 with saturated sodium bicarbonate or hydrochloric acid, then 1 mL of the intestinal phase juice was added, and the subsequent operations were the same as the previous step.

Bioaccessibility (BA) is calculated according to Equation (2).
(2)$$\mathrm{BA}\%=\frac{C\;\times \;V}{M\;\times \;\mathrm{TC}}\;\times \;100\%$$
where BA is the bioaccessibility of metals in the human digestive system (%), C is the soluble content of metals in the reaction solution in the in vitro gastric or intestinal phase (mg/L), V is the volume of added solution (L), TC is the total concentration of heavy metals (mg/kg), and M is the mass of sample in the centrifuge tube (kg).

### 2.6. Human Health Risk Assessment

#### 2.6.1. Scenarios of Risk Assessment

Two scenarios were simulated to access the health risk caused by ingestion with different particle fractions in children: (I) health risks based on the total Cr content and Cr gastrointestinal bioaccessibility of different particle fractions (0–50 μm, 50–100 μm, 100–250 μm, 250–500 μm, and 500–1000 μm); and (II) health risks conducted with the particle mass (0–50 μm, 0–100 μm, 0–250 μm, 0–500 μm, and 0–1000 μm) and metal content (total Cr content and Cr bioaccessibility) in different particle fractions. In Scenario II, 0–100 μm soil particles were the sum of 0–50 μm and 50–100 μm particles. Similarly, in the 0–250 μm fraction, Cr content was the sum composed of 0–50 μm, 50–100 μm, and 100–250 μm particle concentrations. This study considered children as an example to assess the noncarcinogenicity under the two scenarios.

#### 2.6.2. Noncarcinogenic Health Risk Assessment

Generally, pollutants in the environment enter the human body in three ways: oral–hand contact, breathing inhalation, and dermal contact. Compared to adults, children are more vulnerable and bear greater risks due to their object-to-mouth and hand-to-mouth behaviors and shorter stature and lighter weight, which result in a greater intake of surface soils [32,33,34]. This study considered children to be more susceptible to potential hazards from pollutants because of their higher frequency of hand-to-mouth contact than adults.

The hazard quotient (HQ) was used to evaluate noncarcinogenic risks. In this study, noncarcinogenic risks to children caused by total Cr content and Cr bioaccessibility in different particle fractions were calculated under two scenarios. An HQ less than 1 indicated that the pollutant was unlikely to harm the human body, but a potential noncarcinogenic risk may occur if the HQ is more than 1. Note that the HQ is calculated according to Equation (3) [17].
(3)$$\mathrm{HQ}=\frac{\mathrm{ADD}}{\mathrm{RfD}}$$
where ADD is the average daily exposure dose (mg/d/kg), and RfD is the daily reference intake (mg/d/kg) for humans. The oral RfD values for Cr were set at 1.50 mg/d/kg [35].

In this study, the average daily exposure dose (ADD) of metal elements in samples that entered the children’s bodies through oral–hand contact was calculated according to the equation from the Exposure Factor Manual [36]. The exposure dose calculation based on the total Cr content and Cr bioaccessibility was performed according to Equations (4) and (5).
(4)$${\mathrm{ADD}}_{\mathrm{oral-soil}}=\frac{C_{s}{\;\times \;\mathrm{IR}}_{s\;}\times \;\mathrm{CF}\;\times \;\mathrm{EF}\;\times \;\mathrm{ED}}{\mathrm{BW}\;\times \;\mathrm{AT}}$$
(5)$${\mathrm{ADD}}_{\mathrm{oral-soil}}=\frac{C_{s}{\;\times \;\mathrm{IR}}_{s\;}\times \;\mathrm{CF}\;\times \;\mathrm{EF}\;\times \;\mathrm{ED}\;\times \;\mathrm{BA}}{\mathrm{BW}\;\times \;\mathrm{AT}}$$
where C_s_ is the concentration of pollutants in different particle fractions (mg/kg); IRs is the ingestion rate (mg/d), which was assumed to be 200 mg/d for children [37]; CF is the quality conversion factor, 1 × 10^−3^ (L/cm^3^); EF is the exposure frequency (day/year) and was assumed at 350 day/year; ED is the exposure duration (year), which was assumed to be 6 years for children in assessing HQ [38]; BW is body weight (kg), which was assumed to be 17.1 kg for children [39]; and BA is the Cr gastrointestinal bioaccessibility of the two methods in this study.

### 2.7. Quality Assurance and Quality Control

All laboratory glassware was immersed in a 10% nitric acid solution for 24 h before use and then rinsed with deionized water three times. All experiments were conducted in triplicate, except for the in vitro bioaccessibility tests, which were carried out in duplicate. All the chemicals used in the experiment were of analytical grade, and ultrapure water was used throughout the whole experiment. For each experiment batch, two or three blank samples were included for quality control.

The relative standard deviation (RSD) ranged from 1.85 to 6.90% for total Cr in soil samples in all particle sizes. In the gastric phase of bioaccessibility tests, the RSD was 1.14–5.32% and 3.60–6.63% for the SBRC method and PBET method, respectively. For the gastrointestinal phase, the RSD was 3.65–8.74% and 1.70–5.90% for the SBRC method and PBET method, respectively. The RSD results indicated a high level of agreement between replicate experimental runs and satisfactory reproducibility of the PBET method and SBRC method.

### 2.8. Statistical Analysis

All experiment results in this study are expressed as mean ± standard deviation. All data were processed using Microsoft Excel 2016 (Microsoft Corporation, Redmond, WA, USA) and IBM SPSS Statistics 24.0 (IBM Corporation, Armonk, NY, USA). Duncan analysis was used to conduct the significant difference test of data at the 5% level. Pearson correlation coefficient tests were used to investigate the correlation between particle size and other soil physicochemical properties. Data and graphs were processed by Origin 2018 (OriginLab Corporation, Northampton, MA, USA).

## 3. Results and Discussion

### 3.1. Sample Properties in Different Particle Sizes

The mean pH of all size fractions was 5.31, 7.86, and 9.53 for Samples L, M, and H, respectively. Sample L was acidic (range: 5.15–5.63), and the others were alkaline (range: 7.71–9.67) (Table 1). No significant correlation was found between particle size and pH (r = 0.377, −0.310, 0.283; *p* > 0.05) (Appendix A). The mean Fe content was 3342.3, 31,613.0, and 2031.1 mg/kg for Samples L, M, and H and was concentrated in the 0–50 μm fraction for Samples M (45,029.0 mg/kg) and H (2254.5 mg/kg) (Table 1). The 0–50 μm particle size was significantly and negatively correlated with Fe (r = −0.928, −0.680, −0.632, *p* < 0.05) (Appendix A), suggesting that Fe tended to accumulate in fine particles.

The organic matter (OM) content tended to increase first and then decreased with the decreasing particle size, reaching the maximum value at 50–100 μm. The highest OM content was in Sample H (760.8 mg/kg) compared to Samples M (671.8 mg/kg) and L (561.9 mg/kg) (Table 1). The CEC content in Sample H followed a similar pattern and was first increased and then decreased as the particle size decreased (r = −0.518, *p* < 0.05) (Appendix A), reaching the highest in the 100–250 μm fraction (10.4 cmol^+^/kg) (Table 1). The OM content was significantly and positively correlated with CEC (r = −0.639, −0.919; *p* < 0.05) in Samples M and H but not significantly in L (r = −0.285; *p* > 0.05) (Appendix A). The results of the OM content and CEC content were consistent with the results of van der Kallen et al. [8], who reported a similar trend that organic matter content would help increase CEC.

Considering the 0–1000 μm fraction as the bulk sample (i.e., 100%), the mean proportion of particle mass compared to the total dry sample mass was 11.8%, 8.7%, 27.7%, 17.8%, and 34.2% for size fractions of 0–50 μm, 50–100 μm, 100–250 μm, 250–500 μm, and 500–1000 μm, respectively. The main fractions of all samples were 100–250 μm (range: 20.6–32.8%) and 500–1000 μm (range: 22.6–31.4%) (Figure 1a).

The mean values of Cr content were 282.2, 1140.2, and 16,943.6 mg/kg for Samples L, M, and H, respectively (Figure 1b). The local background concentration of 56.5 mg/kg was exceeded by 3.86–374.7 times [40], indicating the existence of Cr pollution in this area. The Cr content in Sample H increased first and then decreased with the decreasing particle size and reached maximum at the 100–250 μm fraction (21,184.1 mg/kg), but it increased steadily as the particle size decreased and reached maximum at the 0–50 μm fraction for Samples L (374.7 mg/kg) and M (1730.7 mg/kg) (Figure 1b). The distributional pattern of Cr in Samples L and M was consistent with the results of Li et al. [41] and Yutong et al. [42], showing a similar distribution of heavy metals in the fine particle fraction.

The largest AF value was 1.40 and 1.93 for Samples L and M in the 0–50 μm fraction and 1.21 for Sample H in the 100–250 μm fraction (Figure 1c). Moreover, the AF value of the 50–100 μm fraction for all samples was larger than 1, indicating that Cr tended to accumulate in the finer sizes (0–100 μm), which was similar to the results of Zhang et al. [43].

The higher Cr content in Sample H may be related to the higher pH, higher CEC content, higher OM content, and lower Fe content obtained in Sample H than L and M. The results were consistent with the results of Li et al. [41] and Shahid et al. [44]. At low pH (pH < 6), competition by hydrogen ions for binding sites will enhance the release of heavy metals from soil binding sites into soil solutions. At high pH (pH > 8), there are no protons to compete for binding sites, so solid-phase exchange sites can bind to metal cations or allow for precipitation to occur in the soils [44,45], which explains how a high soil pH may increase the metal content in the soils. Additionally, Fe can compete with other metal ions for adsorption sites in the soil through mechanisms such as ion exchange, adsorption, and surface precipitation, thereby reducing the adsorption of Cr. Moreover, high CEC content would likely enhance the adsorption of heavy metals in soils [46]. OM acts as a carrier of Cr via binding, and high OM content would also enhance the Cr content in soils [47]. The impact of Fe, CEC, and OM on Cr indicated that lower Fe content, higher CEC, and lower OM content would lead to higher Cr content.

### 3.2. Bioaccessibility of Cr in Different Particle Size Ranges

#### 3.2.1. Bioaccessible Concentration of Cr

The extraction content in the gastric phase of the SBRC method was higher than that of the PBET method, suggesting that the SBRC method may have a stronger extraction capacity for Cr, which may be due to the lower pH (1.5) of the SBRC method [48]. The possible reason for this is that the acidic environment promotes the dissolution of Cr(III) in chromium-bearing minerals, thereby increasing the Cr BA (%) values in the G phase [49]. The Cr BAC (mg/kg) value was 880.6–9380.3 mg/kg, 15.0–116.4 mg/kg, and 19.8–91.5 mg/kg for Samples L, M, and H (Table 2), respectively, which was consistent with the variations in the Cr content in different samples. However, Cr BA (%) in Sample L was 0.70–30.9% higher than in Samples M and H (Figure 2), suggesting that the bioaccessibility of Cr in the low Cr-polluted soil was greater than in the high Cr-polluted soil to some extent.

#### 3.2.2. Bioaccessibility of Cr

Cr BA (%) in Samples L and M tended to decrease as the particle size decreased (Figure 2). Cr BA (%) estimated by the SBRC method in Sample H peaked at the 50–100 μm particle sizes, with 52.2% and 20.9% for the G phase and GI phase, respectively (Figure 2). For the PBET method, Cr BA (%) in Sample H reached maximum in the 0–50 μm fraction, with 13.4% and 11.4% for the G phase and GI phase, respectively (Figure 2). Cr BA (%) of all samples was significantly negatively correlated with the total Cr content (r = −0.525~−0.943, *p* < 0.05) (Appendix A). The results showed that the distributional pattern of Cr BA (%) was similar to that of the total Cr content, suggesting the bioaccessibility may be related to the total Cr content [50]. Meanwhile, the low BA (%) of smaller particles may be due to the greater adsorption capacity and greater surface area for certain metals of fine particles, thus reducing the BA (%) of smaller particle sizes [8].

In the GI phase, the Cr BA (%) values of the SBRC method ranged from 0.84 to 37.5%, decreasing by an average of 128.5% compared to the G phase (Figure 2). In addition, the Cr GI BA (%) values of the PBET method ranged from 2.04 to 21.7%, with a mean enhancement of 75.4% for Samples L and M and a mean reduction of 2.35% for Sample H from the G phase to GI phase (Figure 2).

The decreasing bioaccessibility may be related to lower pH in the G phase compared to the GI phase, which may contribute to the dissolution of heavy metals [17]. Additionally, Cr and Fe could form coprecipitation due to the increase in pH, resulting in lower BA (%) in the GI phase. Moreover, soil organic matter and Fe can increase the fixation of metals in soils through adsorption reactions, resulting in bioaccessibility reduction [51,52]. The possible reason for increasing BA (%) was that Cr is easier to transform into Cr(VI) in the alkaline environment during the gastrointestinal phase, which is stable and has strong mobility in a weakly alkaline condition [53]. Other studies have shown that the pepsin contains multiple −COOH and −NH_2_, and these groups can interact with Cr to form a complex, thereby reducing the Cr BA (%) values in the G phase [54,55]. In the GI phase, bile could promote the dissolution of heavy metals by reducing the interfacial tension and complexation of complexes from the G phase [56], and Cr can be easily released in the digestive juice that helps increase Cr BA (%) in the GI phase.

### 3.3. Health Risk Contribution in Different Particle Size Ranges

#### 3.3.1. Health Risks

The distributional pattern of HQ is consistent with Cr content and Cr bioaccessibility. The noncarcinogenic risk of all samples was less than 1 under the two scenarios, which was within the tolerance level (HQ = 1) for children, indicating that Cr has no potential health risk to children (Figure 3).

In Scenario I, the HQ values based on the total Cr content (i.e., TC-I) were 1.67–114.4 times higher than those based on the bioaccessible Cr content (i.e., SBRC-I, PBET-I). The results indicated that the HQ based on bioaccessibility was one to two orders of magnitude less than that based on total Cr concentration, which would be closer to the actual situation rather than the total Cr content. In Scenario II, the HQ values conducted with the particle mass generally declined 0.50–42.7% compared to Scenario I but increased by 0.10–80.6% in some cases.

The results showed that the HQ of the 50–100 μm fraction (Figure 3I) was overestimated when associated with particle mass, while the HQ of the 250–500 μm and 500–1000 μm fractions was underestimated, indicating that the health risk of a single particle range may not truly reflect the reality. The health risk result was less severe in Scenario II than in Scenario I, which tended to distribute the risk evenly when taking particle mass into account.

#### 3.3.2. Risk Contribution of Different Particles

The combined effect of the metal content and the particle mass on HQ was shown as a percentage (%) to quantify the degree of influence. The 50–100 μm and 100–250 μm fractions contributed the main proportion of the health risk based on metal content. The contribution of 50–100 μm fraction was 23–29%, 27–30%, and 20–22% for Samples L, M, and H, respectively (Figure 4a). The contribution of the 100–250 μm fraction was 18–24%, 17–24%, and 20–25% for Samples L, M, and H, respectively (Figure 4a). However, the fractions that contributed mostly to health risk changed when the health risk was conducted with the combined effect of metal content and particle mass of the 100–250 μm and 500–1000 μm fractions. The 100–250 μm and 500–1000 μm fractions contributed the main proportion to health risk based on the combined effect. The contribution of the 100–250 μm fraction was 31–38%, 22–24%, and 31–36% for Samples L, M, and H, respectively (Figure 4b). The contribution of the 500–1000 μm fraction was 19–23%, 37–46%, and 28–30% for Samples L, M, and H, respectively (Figure 4b).

The proportion of the health risk from particle size fractions declined when considering the combined effect and was 2.40–16.3%, 9.20–18.9%, and 0.30–4.8% for the 0–50 μm, 50–100 μm, and 250–500 μm fractions, respectively, while it increased by 3.10–15.1% and 1.70–26.3% for the 100–250 μm and 500–1000 μm fractions. The possible reason for the increasing proportion of the health risk may be due to the large particle mass proportion of the 100–250 μm and 500–1000 μm fractions, which was 27.7% and 34.2% on average, respectively. The mean proportion of the 100–250 μm fraction was 57.6%, 68.8%, and 35.9% higher than that of the 0–50 μm, 50–100 μm, and 250–500 μm fractions, respectively, while the mean proportion of the 500–1000 μm fraction was 65.6%, 74.7%, and 48.0% higher than the 0–50 μm, 50–100 μm, 250–500 μm fractions, respectively. The result showed that the particle fraction with a higher mass proportion may bring out higher risks to some extent. Deviation occurs when health risk is based on a specific particle size, which may reduce the precision of the health risk to the human body from a heterogeneous body such as soils or sediments.

## 4. Conclusions

The influence of soil particle size on Cr content, physicochemical properties distribution, and metal bioaccessibility was investigated, and the impact of soil particle size and mass distribution on human health risks was also examined. Higher Cr content may be related to higher pH, higher CEC content, higher OM content, and lower Fe content. Cr tended to accumulate in the finer soil fractions (0–100 μm). The SBRC method may have a stronger extraction capacity for Cr, which may be due to the lower pH that promotes the dissolution of Cr(III) in chromium-bearing minerals, thereby increasing the Cr BA (%) values. The gastrointestinal bioaccessibility of the less Cr-polluted soils (Samples L and M) decreased as the particle size decreased but increased in the highly Cr-polluted soil (H). Higher bioaccessibility (0.70–30.9%) in Sample L than in other soils indicated that the mobility of Cr in the less-Cr-polluted soil was greater than in the more Cr-polluted soils to some extent. The main proportion of the Cr health risk contribution was in the 50–100 μm and 100–250 μm fractions for health risk based on metal content, which was 22%–30% and 17–25%, respectively. However, the dominant fractions changed in the 100–250 μm and 500–1000 μm fractions when health risk was conducted with the combined effect of metal content and particle mass, which was 22–38% and 19–46%, respectively.

## Figures and Tables

**Figure 1 ijerph-19-05253-f001:**
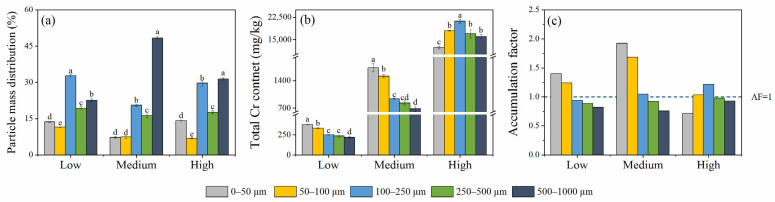
Particle mass distribution of all samples (**a**). Total Cr content of all samples (**b**). Accumulation factor of Cr of all samples (**c**). The error bars represent standard deviation. Means with different letters in the same row are significantly different at *p* < 0.05.

**Figure 2 ijerph-19-05253-f002:**
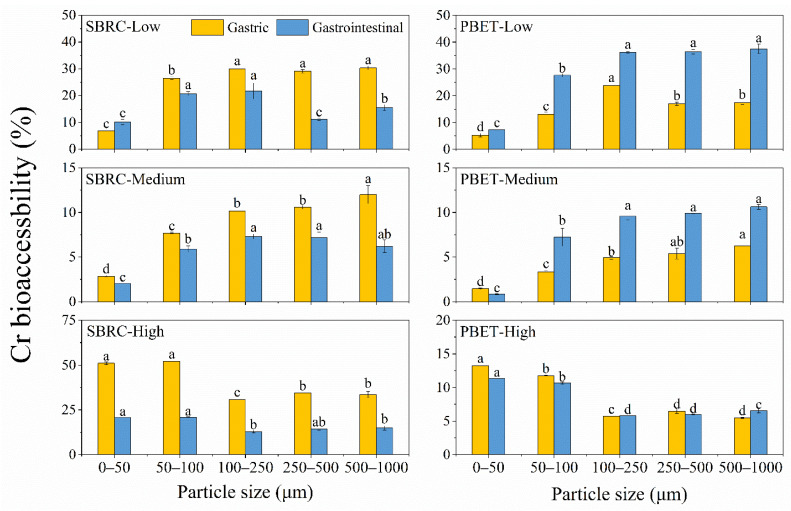
Cr bioaccessibility of different fractions estimated through the PBET and SBRC methods. All values are presented as mean ± standard deviation. The error bars represent standard deviation. Means with different letters in the same row are significantly different at *p* < 0.05.

**Figure 3 ijerph-19-05253-f003:**
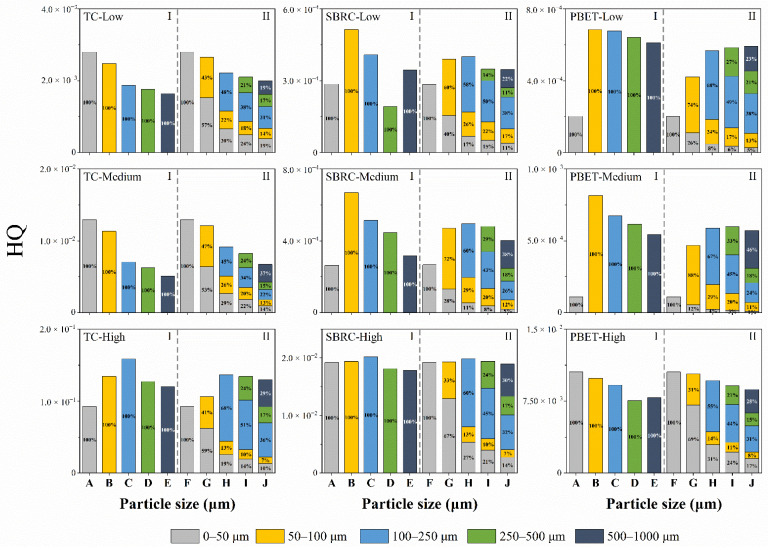
Cr hazard quotient and the particle mass contribution of Cr hazard quotient of children over different particle size fractions. HQ: hazard quotient; TC: total Cr content; SBRC: Cr bioaccessible concentration estimated through the SBRC method; PBET: Cr bioaccessible concentration estimated through the PBET method; I: Scenario I; II: Scenario II; A: 0–50 μm; B: 50–100 μm; C: 100–250 μm; D: 250–500 μm; E: 500–1000 μm; F: 0–50 μm; G: 0–100 μm; H: 0–250 μm; I: 0–500 μm; and J: 0–1000 μm.

**Figure 4 ijerph-19-05253-f004:**
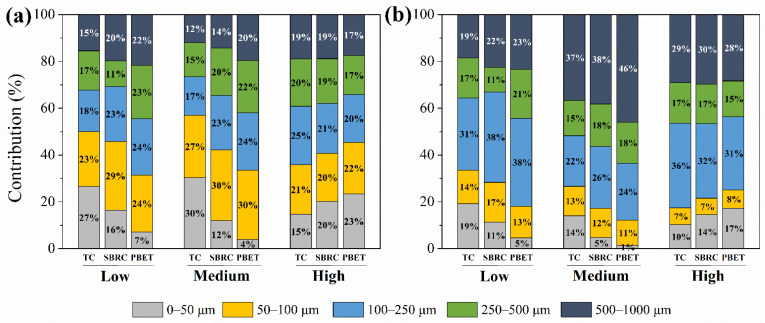
Contribution of different particle size fractions to Cr hazard quotient. Health risk based on (**a**) metal content and (**b**) combined effect of metal content and particle mass. TC: total Cr content; SBRC: Cr bioaccessible concentration estimated through the SBRC method; PBET: Cr bioaccessible concentration estimated through the PBET method.

**Table 1 ijerph-19-05253-t001:** Physicochemical properties of various particle size fractions.

Soil Properties	Particle Fraction (μm)				
	0–50	50–100	100–250	250–500	500–1000
Low					
pH	4.95 ± 0.06 a	5.09 ± 0.26 a	5.16 ± 0.03 a	5.21 ± 0.05 a	5.65 ± 0.57 a
OM (mg·kg^−1^)	508.68 ± 34.56 ab	570.85 ± 42.83 a	504.53 ± 71.45 ab	545.44 ± 65.59 ab	463.44 ± 46.8 b
CEC (cmol^+^·kg^−1^)	1.65 ± 0.27 a	1.29 ± 0.13 b	1.14 ± 0.02 b	1.38 ± 0.07 ab	1.21 ± 0.04 b
Total Fe (mg·kg^−1^)	4540.58 ± 284.49 a	4691.3 ± 299.08 a	3146.38 ± 172.68 b	2317.39 ± 492.75 c	2015.94 ± 65.27 c
Medium					
pH	8.35 ± 0.09 a	7.71 ± 0.11 a	7.86 ± 0.45 a	7.71 ± 0.51 a	7.94 ± 0.39 a
OM (mg·kg^−1^)	649.15 ± 24.42 ab	677.18 ± 49.86 a	597.34 ± 27.31 bc	613.38 ± 24.66 bc	585.27 ± 19.57 c
CEC (cmol^+^·kg^−1^)	3.41 ± 0.13 a	2.14 ± 0.25 b	0.83 ± 0.16 c	0.28 ± 0.06 d	0.14 ± 0.01 d
Total Fe (mg·kg^−1^)	45028.99 ± 652.66 a	37537.97 ± 1383.63 b	21666.67 ± 3263.28 d	22616.23 ± 2826.12 d	30348.41 ± 4589.14 c
High					
pH	9.5 ± 0.04 a	9.37 ± 0.18 a	9.67 ± 0.06 a	9.46 ± 0.12 a	9.63 ± 0.27 a
OM (mg·kg^−1^)	741.33 ± 15.03 a	760.8 ± 10.74 a	680.06 ± 16.44 b	658.8 ± 8.99 b	601.49 ± 19.79 c
CEC (cmol^+^·kg^−1^)	9.66 ± 0.13 b	9.71 ± 0.18 b	10.45 ± 0.12 a	9.83 ± 0.03 b	8.23 ± 0.29 c
Total Fe (mg·kg^−1^)	2254.48 ± 236.38 a	2102.61 ± 286.65 a	2152.06 ± 327.93 a	1818.92 ± 40.96 a	1827.54 ± 65.27 a

All values are presented as mean ± standard deviation. Means with different letters in the same row are significantly different at *p* < 0.05; OM: organic matter; CEC: cation exchange capacity.

**Table 2 ijerph-19-05253-t002:** Bioaccessible Cr concentrations in the simulated gastrointestinal solutions based on the physiologically based extraction test (PBET) and Solubility Bioaccessibility Research Consortium (SBRC) methods for the three samples.

Element (mg·kg^−1^)	Particle Fraction (μm)				
	0–50	50–100	100–250	250–500	500–1000
Low					
PBET-G	19.78 ± 2.51 c	43.23 ± 2.97 b	59.43 ± 1.58 a	40.1 ± 1.46 b	38.04 ± 1.45 b
PBET-GI	27.27 ± 0.45 c	91.46 ± 1.94 a	90.55 ± 0.71 a	85.91 ± 1.94 ab	81.83 ± 3.83 b
SBRC-G	25.92 ± 0.32 d	87.93 ± 1.75 a	75.19 ± 0.85 b	68.7 ± 1.65 c	66.36 ± 1.64 c
SBRC-GI	38.19 ± 3.98 c	68.76 ± 2.57 a	54.5 ± 7.44 b	26.53 ± 1.05 d	34.05 ± 2.43 cd
Medium					
PBET-G	25.23 ± 1.3 b	50.66 ± 1.52 a	46.39 ± 1.5 a	44.73 ± 5.08 a	42.4 ± 4.72 a
PBET-GI	14.99 ± 0.46 c	114.62 ± 7.61 a	91.45 ± 1.96 b	82.07 ± 0.35 b	73.12 ± 0.93 b
SBRC-G	49.54 ± 1.3 d	116.41 ± 1.41 a	95.41 ± 1.78 b	87.93 ± 1.75 bc	81.84 ± 6.87 c
SBRC-GI	35.44 ± 0.98 c	89.29 ± 5.4 a	68.76 ± 2.57 b	59.78 ± 5.03 b	42.39 ± 4.8 c
High					
PBET-G	1669.38 ± 30.11 b	2112.26 ± 19.46 a	1204.97 ± 68.28 c	1098.97 ± 53.74 c	880.58 ± 12.02 d
PBET-GI	1410.81 ± 1.91 b	1907.94 ± 15.62 a	1203.99 ± 32.07 c	1018.39 ± 10.02 d	1068.91 ± 25.73 d
SBRC-G	6340.63 ± 96.01 b	9380.3 ± 139.9 a	6525.62 ± 27.58 b	5861.86 ± 12.02 c	5411.21 ± 275.79 d
SBRC-GI	2573.38 ± 45.05 bc	3752.62 ± 43.5 a	2714.97 ± 145.17 b	2428.25 ± 90.11 bc	2389.31 ± 185.22 c

All values are presented as mean ± standard deviation. Means with different letters in the same row are significantly different at *p* < 0.05. “G”, gastric phase; “GI”, gastrointestinal phase.

## Data Availability

Not applicable.

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
