# Peer review of "Relative Contribution of Metal Content and Soil Particle Mass to Health Risk of Chromium-Contaminated Soil"

_ijerph, 2022, doi:10.3390/ijerph19095253_

Round 1

Reviewer 1 Report

In general terms the study is interesting, here are some suggestions to improve your text.
In terms of human toxicity, not only the particle size, but also the stages of trivalent and hexavalent chromium should be considered in this article, at least in the introduction, there is a vast amount of published articles explaining this concept, they need to expand their background.
They need to explain if the oxidation number of chromium is not affected by the chemical environment of their gastrointestinal model both in the results and in their final conclusions.
It is desirable that they use an outline of the method they explain with the text on lines 123-153.

Reviewer 2 Report

please see attached file.

Round 2

Reviewer 2 Report

There is still the issue of consistency in listing numbers and units throughout the manuscript.  For all units, whether concentration, mass, volume, etc., either use the style:

###unit such as 111ug/L or

unit such as 111 ug/L

The exception could be made for % where it would be 55% not 55 %.

This discrepancy starts right in the abstract and continues through document.  Line 18 has 15.87% while line 22 has 43.46 %.
